# Outcomes of acute coronary syndrome patients with concurrent extra-cardiac vascular disease in the era of transradial coronary intervention: A retrospective multicenter cohort study

Masaki Kodaira[1]*, Mitsuaki Sawano[2], Toshiki Kuno[1,3], Yohei Numasawa[1], Shigetaka Noma[4], Masahiro Suzuki[5], Shohei Imaeda[2], Ikuko Ueda[2], Keiichi Fukuda[2], Shun Kohsaka[2]

1 Department of Cardiology, Japanese Red Cross Ashikaga Hospital, Tochigi, Japan, 2 Department of Cardiology, Keio University School of Medicine, Tokyo, Japan, 3 Department of Medicine, Icahn School of Medicine at Mount Sinai, Mount Sinai Beth Israel, New York, New York, United States of America, 4 Department of Cardiology, Saiseikai Utsunomiya Hospital, Tochigi, Japan, 5 Department of Cardiology, National Hospital Organization Saitama Hospital, Saitama, Japan

* m.kodaira@ashikaga.jrc.or.jp

## Abstract

### Background

Extra-cardiac vascular diseases (ECVDs), such as cerebrovascular disease (CVD) or peripheral arterial disease (PAD), are frequently observed among patients with acute coronary syndrome (ACS). However, it is not clear how these conditions affect patient outcomes in the era of transradial coronary intervention (TRI).

### Methods and results

Among 7,980 patients with ACS whose data were extracted from the multicenter Japanese percutaneous coronary intervention (PCI) registry between August 2008 and March 2017, 888 (11.1%) had one concurrent ECVD (either PAD [345 patients: 4.3%] or CVD [543 patients; 6.8%]), while 87 patients (1.1%) had both PAD and CVD. Overall, the presence of ECVD was associated with a higher risk of mortality (odds ratio [OR]: 1.728; 95% confidence interval [CI]: 1.183–2.524) and bleeding complications (OR: 1.430; 95% CI: 1.028–2.004). There was evidence of interaction between ECVD severity and procedural access site on bleeding complication on the additive scale (relative excess risk due to interaction: 0.669, 95% CI: -0.563–1.900) and on the multiplicative scale (OR: 2.105; 95% CI: 1.075–4.122). While the incidence of death among patients with ECVD remained constant during the study period, bleeding complications among patients with ECVD rapidly decreased from 2015 to 2017, in association with the increasing number of TRI.

### Conclusions

Overall, the presence of ECVD was a risk factor for adverse outcomes after PCI for ACS, both mortality and bleeding complications. In the most recent years, the incidence of

**Data Availability Statement:** Data are from the Japan Cardiovascular Database-Keio interhospital

Cardiovascular Studies whose authors may be contacted at hqa-adm@umin.ac.jp. Data are available only upon request according to the "Act on the Protection of Personal Information" Law (as of May 2017) and the "Ethical Guidelines for Medical and Health Research Involving Human Subjects" (as of March 2015). The current study data was obtained from the JCD-KiCS PCI registry and would be available upon request to University of Tokyo, Healthcare Quality Assessment. (E-mail: hqa-adm@umin.ac.jp.)

**Funding:** This research was funded by a grant from the Ministry of Education, Culture, Sports, Science and Technology, Japan (KAKENHI No. 21790751, 16H05215).

**Competing interests:** Dr. Kohsaka has received grants from Bayer Yakuhin and Daiichi-Sankyo; has received lecture fees from Bayer Y.akuhin and Bristol-Myers Squibb. This does not alter our adherence to PLOS ONE policies on sharing data and materials.

bleeding complications among patients with ECVD decreased significantly coinciding with the rapid increase of TRI.

## Introduction

Recent advancements in percutaneous coronary intervention (PCI) devices and techniques, such as the use of drug-eluting stents with improved safety and efficacy, has enabled interventionists to perform PCI in high-risk patients, including those with concurrent extra-cardiac vascular disease (ECVD).[1] This condition has been referred to as the presence of cerebrovascular disease (CVD) or peripheral artery disease (PAD) in addition to established coronary artery disease.[2–4] Indeed, performing PCI in patients with concurrent ECVD has been reported to be associated with lower procedural success rates and higher complication rates, [1] as demonstrated by the Global Registry of Acute Coronary Events (GRACE)[5] and the Reduction of Atherothrombosis for Continued Health Registry.[6] Previous studies from Europe and the United States have shown the negative prognostic value of ECVD for both short- and long-term outcomes of patients receiving PCI.[7–10] In addition, a recent study of patients with acute coronary syndrome (ACS) from Italy reported that the risk of mortality at 5 years post-PCI doubled for patients with one ECVD (62% among patients with CVD and 63% among patients with PAD) and increased by a further 80% in patients with both CVD and PAD, compared with a 33% mortality rate for ACS patients without ECVD.[8] Hence, there is a considerable need to establish an effective therapeutic strategy to improve outcomes of high-risk patients after PCI.

During the last decade, the implementation of strategies to minimize PCI-related bleeding, including transradial intervention (TRI), has been proven to reduce the overall rate of PCI-related bleeding complications.[11, 12] The Minimizing Adverse Haemorrhagic Events by TRansradial Access Site and Systemic Implementation of Angiox (MATRIX) randomized trial reported that the use of TRI was related to reduced adverse clinical events in ACS patients compared to transfemoral intervention (TFI).[13] However, it is unclear whether the modern application of TRI could improve outcomes among patients with ACS and ECVD.[14] With TRI being less invasive than transfemoral intervention (TFI), we hypothesized that TRI may modify the negative effects of ECVD on ACS patients' outcomes; thus, consideration of the interaction of procedural site and severity of ECVD on in-hospital outcomes may influence decision making for high-risk patients. In light of the increasing interest in early interventional strategies for high-risk patients with ACS, establishing and quantifying the contributions of TRI to the health of patients with ACS and concurrent ECVD is of utmost importance. In this study, our primary aim was to examine the impact effect of ECVD (CVD and/or PAD) on in-hospital outcomes among patients with ACS undergoing PCI. Our secondary aim was to assess temporal trends of ACS patients with ECVD and the incidence of in-hospital outcomes, in relation to the increased utilization of TRI in Japan.

## Methods and materials

### Study population and study design

Our study cohort was derived from the Japan Cardiovascular Database-Keio interhospital Cardiovascular Studies (JCD-KiCS) registry, which is an ongoing, prospective, multicenter cohort study designed to collect data on the demographics, procedural characteristics, and outcomes of patients undergoing PCI. Details of the registry have been previously published.[15–17] In

brief, the JCD-KiCS registry collects data on more than 200 variables in accordance with the National Cardiovascular Data Registry CathPCI version 4, the largest registry of PCI in the United States.[18] The JCD-KiCS registry study protocol was approved by the Institutional Review Board of Keio University School of Medicine, as well as those of each participating hospital. The study was carried out in accordance with the approved guidelines and the Declaration of Helsinki. All participants provided informed consent.

To fulfill the purpose of our study and evaluate temporal trends, we selected four hospitals that continually registered patient data from August 2008 to March 2017. Within this selected cohort, a total of 9,209 patients underwent PCI for ACS. We excluded 778 patients who were admitted with cardiogenic shock and 110 patients in cardiopulmonary arrest. Furthermore, we excluded 351 patients due to missing data on patient age and sex, leaving a final cohort of 7,980 patients with ACS (Fig 1). Our primary hypothesis was that ECVD is associated with an increased incidence of adverse clinical outcomes in a cumulative manner, with higher risks observed among patients with ACS and more than two concurrent ECVDs. The secondary hypothesis was that the incidence of adverse in-hospital clinical outcomes has decreased among patients with ECVD in recent years, with a significant association between outcomes and higher utilization of TRI. Therefore, we compared the primary and secondary outcomes between TRI and TFI in the study cohort over the 9-year period, from 2008 to 2017. In addition, we assessed the joint association of ECVD severity and PCI procedural site with in-hospital outcomes, applying both multiplicative interaction and additive interaction analyses.

## Study definitions and outcomes

We considered ECVD lesions to be present when either CVD or PAD was documented prior to or during hospitalization for PCI. Specifically, CVD was defined as a history of stroke or transient ischemic attack, detection of carotid artery stenosis (>79%) via a non-invasive or invasive carotid test, or previous history of carotid artery surgery/intervention for carotid artery stenosis. Peripheral arteries included the aortoiliac, femoral-popliteal, renal, mesenteric, and abdominal aortic arteries. The definition of PAD was taken as claudication; amputation or arterial vascular insufficiency; vascular construction, bypass surgery, or percutaneous intervention to the extremities; documented aortic aneurysm; a positive non-invasive test result (ankle brachial index (ABI) $\leq$ 0.9); and ultrasound, magnetic resonance imaging, computed tomography, or angiographic imaging indicating >50% stenosis in any peripheral artery. The primary outcome of interest in the present analysis was in-hospital death. The secondary outcome was the development of a type 3 bleeding complication, as defined by the Bleeding Academic Research Consortium criteria.[19]

## Statistical analysis

For descriptive analysis, we used Pearson chi-square tests to compare categorical values and Student's t test to compare continuous variables between the groups. All continuous variables are presented as number (percentage) or mean ± standard deviation. First, baseline characteristics and outcomes were stratified and compared according to the number of extra-cardiac atherosclerotic sites. Second, multivariate logistic regression analysis was performed to assess the effect of ECVD after adjusting for potential confounders, including mortality and bleeding events. The multivariate logistic regression models included variables incorporated in the National Cardiovascular Data Registry risk models for in-hospital mortality[20] and bleeding risk,[21] comprising the following: age, ECVD, chronic kidney disease (CKD), ST-elevation myocardial infarction (STEMI), chronic lung disease, heart failure upon admission, and history of heart failure comprising in-hospital mortality and age, sex, body mass index (BMI),

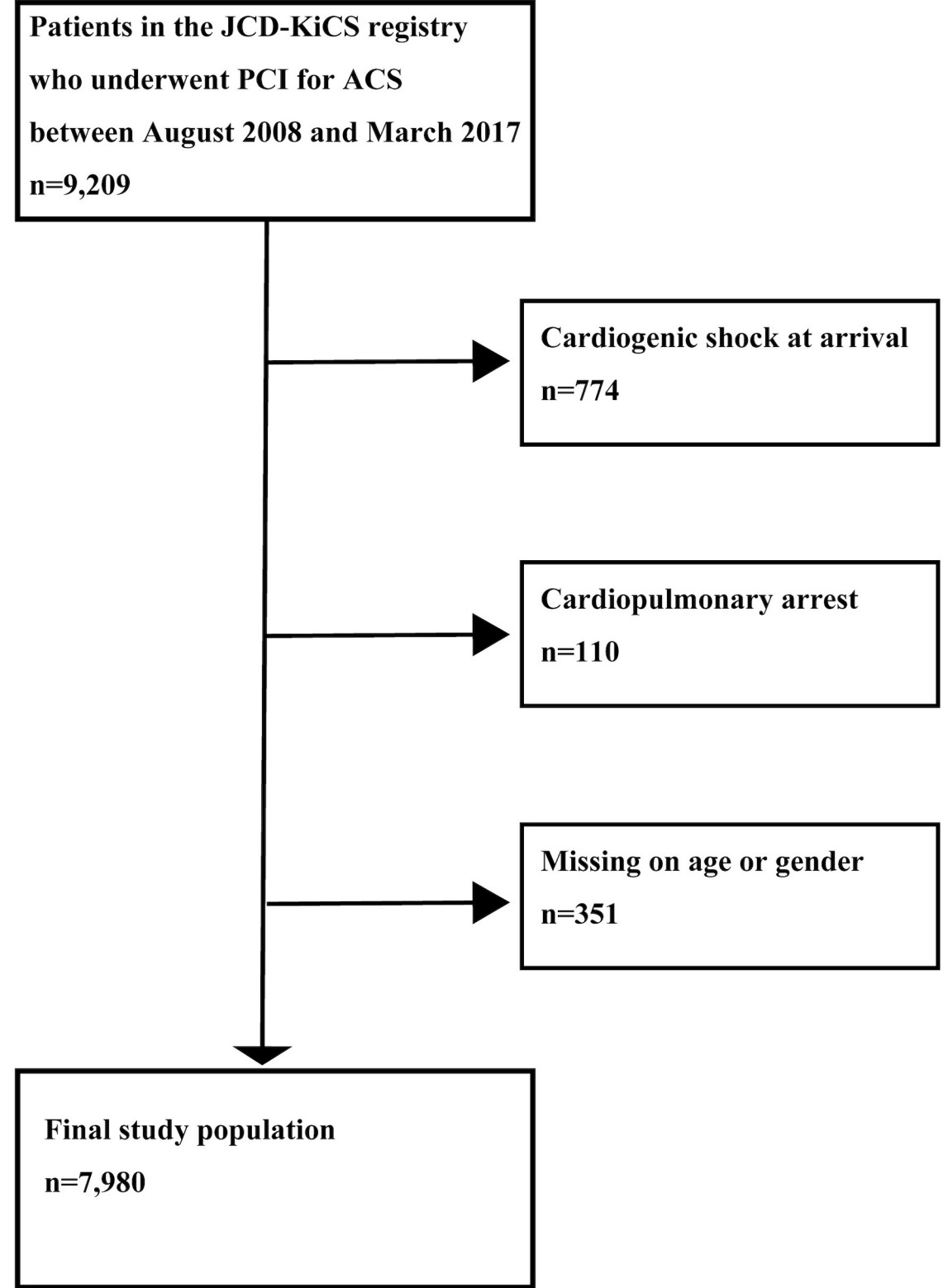

**Fig 1. Study flow chart.** Flow diagram showing the derivation of the final study cohort. JCD-KiCS, Japan Cardiovascular Database-Keio interhospital Cardiovascular Studies; PCI, percutaneous coronary intervention; ACS, acute coronary syndrome.

ECVD, CKD, STEMI, history of PCI, and hemoglobin for bleeding risk. Third, we evaluated whether the association between ECVD severity and in-hospital outcomes among ACS patients differed according to procedure site using multiplicative and additive interaction analyses. Results of interaction were reported according to the recommendations from the International Epidemiological Association.[22] Relative excess risk of interaction (RERI) was used to describe the magnitude of risk due to additive interaction, with a score of >0 taken as evidence of positive additive interaction.[23] Multiplicative interaction was described as an adjusted odds ratio for the interaction term in the logistic regression model, with the variables shown above. Finally, we examined temporal trends in patient characteristics and clinical outcomes over time. We evaluated three consecutive periods, namely, 2008–2010, 2011–2014, and 2015–2017, and included these categories in addition to the above variables to adjust for the registration year in multivariate logistic regression models. Then, we calculated the unadjusted in-hospital mortality and bleeding complications for each period and examined whether trends differed based on the presence of ECVD. In addition, comparison of the primary and secondary endpoints between the two different access sites, TRI and TFI, was performed using similar multivariable regression analyses and covariates. We then calculated the expected outcomes using the previously published regression model. The observed rates of death and bleeding events among patients with ECVD were divided according to their expected rates to obtain the observed/expected (O/E) outcome ratios.[20, 24, 25]

Records with missing data on sex or age (351 patients) were excluded as part of our sampling plan (Fig 1). Consequently, our study population had <0.5% missing data for the covariates used in the death or bleeding complications model, except for estimated glomerular filtration rate (2.6%), pre-procedure hemoglobin level (0.7%), and BMI (6.7%). A single imputation approach was used to handle these missing data. Specifically, missing categorical covariates were set to their lowest risk value, while continuous covariates were replaced with sex-specific medians.

Pre-specified subgroup analyses were conducted for sex (female/male), age groups (age ≥80/age <80), clinical presentation (STEMI/unstable angina or non-STEMI), and access site (TRI/TFI). Statistical analyses were conducted using SPSS version 24.0 (IBM Corp. Armonk, NY, USA) and STATA version 15.1 (StataCorp, College Station, Texas) for performance of RERI. Statistical significance was set at p < 0.05.

## Results

### Sample characteristics

The current analysis included 7,980 patients who underwent PCI for ACS. Of these, 888 patients (11.1%) had one additional vascular disease (CAD +PAD: 345 patients [4.3%], CAD + CVD: 543 patients [6.8%]), whereas 87 patients (1.1%) had all three types (CAD + PAD + CVD) of vascular disease (Fig 2). The baseline characteristics are presented in Table 1. Patients with ECVDs were significantly older with lower BMIs and were more likely to have undergone PCI or coronary artery bypass grafting in the past. In addition, the prevalence of patients with diabetes, chronic kidney failure, on dialysis, and with heart failure upon admission increased as the number of ECVDs increased. Conversely, patients without ECVDs were more likely to be current smokers. Angiographic and procedural data are shown in Table 2. Patients with ECVDs were more likely to have chronic total occlusion, three-vessel disease, or American Heart Association/American College of Cardiology type C lesions. There were no differences among the three groups in terms of the rate of TRI or intra-aortic balloon pump use.

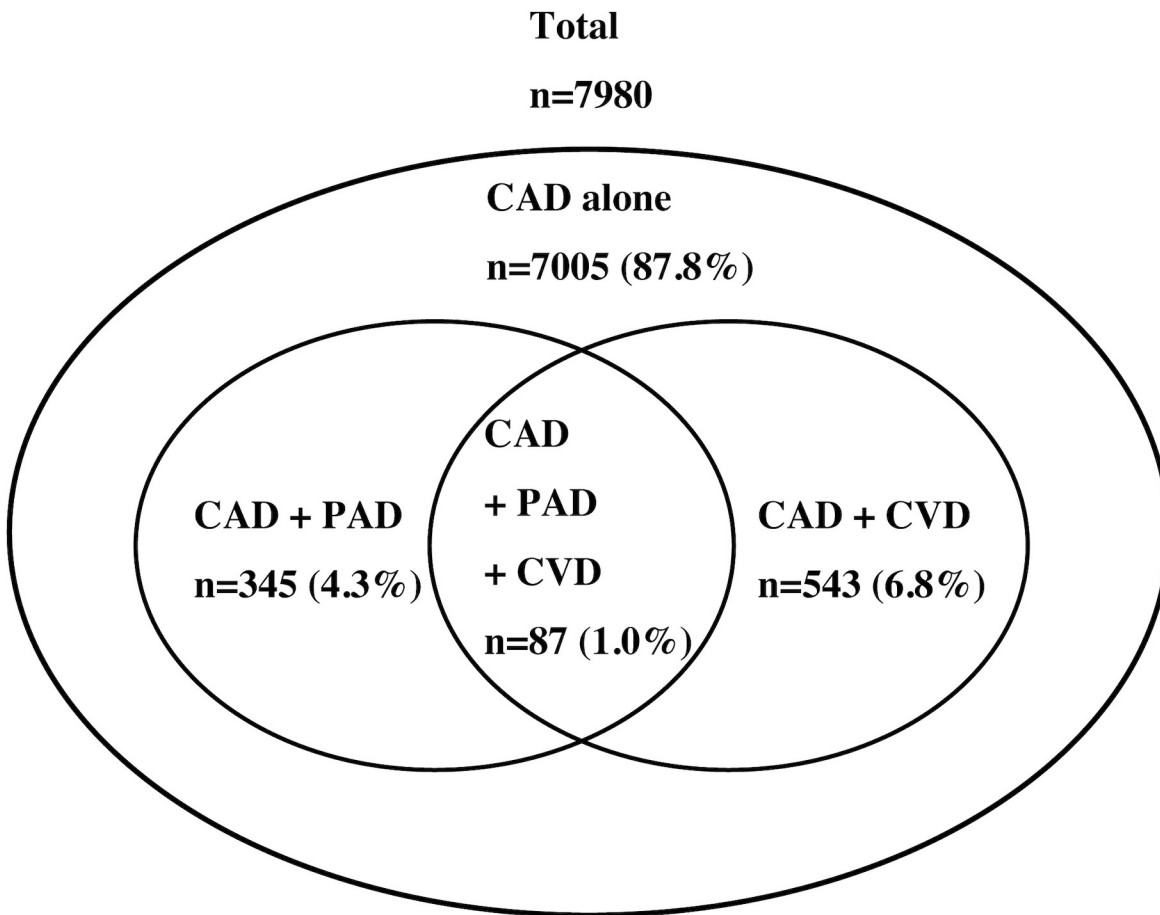

**Fig 2. Distribution of the study population according to the location of extra-cardiac atherosclerosis.** CAD, coronary artery disease; CVD, cerebrovascular disease; PAD, peripheral artery disease.

### Outcomes according to the number of extra-cardiac vascular diseases

The incidence of in-hospital deaths and bleeding complications according to the number of ECVDs is presented in S1 Fig. The risk of in-hospital mortality increased proportionally with the number of additional vascular sites (p < 0.001): CAD alone, 1.8%; CAD and one site, 4.1%; CAD and two sites, 4.6%. The risk of bleeding complications was the highest among patients with a single ECVD compared to those without ECVD or with multiple ECVDs.

After multivariable adjustment, the presence of concurrent ECVD was associated with a higher risk of mortality (odds ratio [OR]: 1.586, 95% confidence interval [CI]: 1.084–2.322) and bleeding complications (OR: 1.460, 95% CI: 1.032–2.064) (Fig 3). The incidence of death increased progressively as the number of concurrent ECVDs increased (OR: 1.457, 95% CI: 1.055–2.012).

### Interaction between extra-cardiac vascular disease and procedural access site

Both multiplicative (OR: 0.864, 95% CI: 0.444–1.680) and additive interaction (RERI: −0.617, 95% CI: −2.432–1.208) showed that ECVD severity and procedural access site had no significant effect on in-hospital mortality (Table 3). On the contrary, interaction was detected

**Table 1. Baseline clinical characteristics in relation with the number of extra-cardiac lesions.**

|  | CAD only n = 7005 | CAD and 1 site n = 888 | CAD and 2 sites n = 87 | p Value |
|---|---|---|---|---|
| Female (%) | 1620 (22.9%) | 198 (21.6%) | 22 (25.3%) | 0.555 |
| Age | 66.8 ± 12.1 | 72.7 ± 10.1 | 74 ± 9.3 | <0.001 |
| BMI | 24.1 ± 3.7 | 23.0 ± 3.6 | 22.7 ± 3.6 | <0.001 |
| History of MI (%) | 1016 (13.9%) | 173 (18.8%) | 16 (18.4%) | <0.001 |
| History of heart failure (%) | 370 (5.1%) | 107 (11.7%) | 16 (18.4%) | <0.001 |
| Diabetes (%) | 2542 (34.7%) | 394 (42.9%) | 46 (52.9%) | <0.001 |
| Dialysis (%) | 203 (2.8%) | 90 (9.8%) | 16 (18.4%) | <0.001 |
| Chronic kidney disease (%) | 2586 (35.3%) | 501 (54.6%) | 62 (71.3%) | <0.001 |
| Chronic lung disease (%) | 188 (2.6%) | 56 (6.1%) | 3 (3.4%) | <0.001 |
| Hypertension (%) | 4925 (67.3%) | 757 (82.5%) | 71 (81.6%) | <0.001 |
| Current smoker (%) | 2741 (37.5%) | 280 (30.5%) | 32 (36.8%) | <0.001 |
| Dyslipidemia (%) | 4393 (60.0%) | 568 (61.9%) | 50 (57.5%) | 0.493 |
| Family history of CAD (%) | 775 (10.6%) | 91 (9.9%) | 8 (9.2%) | 0.218 |
| Atrial fibrillation (%) | 263 (3.6%) | 74 (8.1%) | 10 (11.5%) | <0.001 |
| History of PCI (%) | 1416 (19.4%) | 250 (27.2%) | 26 (29.9%) | <0.001 |
| History of CABG (%) | 199 (2.7%) | 60 (6.5%) | 10 (11.5%) | <0.001 |
| Intervention indication, STEMI (%) | 3194 (45.6%) | 319 (35.9%) | 23 (26.4%) | <0.001 |
| Intervention indication, NSTEMI (%) | 1209 (17.3%) | 180 (20.3%) | 21 (24.1%) | 0.024 |
| Heart failure at admission (%) | 876 (12.0%) | 175 (19.1%) | 22 (25.3%) | <0.001 |

BMI, body mass index; CABG, coronary artery bypass grafting; CAD, coronary artery disease; MI, myocardial infarction; NSTEMI, non-ST-elevation myocardial infarction; PCI, percutaneous coronary intervention; STEMI,ST-elevation myocardial infarction

**Table 2. Procedural characteristics according to the number of extra-cardiac lesions.**

|  | CAD only n = 7005 | CAD and 1 site n = 888 | CAD and 2 sites n = 87 | p Value |
|---|---|---|---|---|
| Procedural characteristics |  |  |  |  |
| Transradial (%) | 2898 (39.6%) | 358 (39.0%) | 26 (29.9%) | 0.175 |
| Transfemoral (%) | 4155 (59.3%) | 500 (56.3%) | 51 (58.6%) | 0.228 |
| IABP (%) | 562 (7.7%) | 82 (8.9%) | 4 (4.6%) | 0.221 |
| Bifurcation (%) | 1877 (27.3%) | 208 (24.1%) | 23 (28.0%) | 0.136 |
| CTO (%) | 152 (6.4%) | 28 (12.2%) | 2 (10.0%) | 0.004 |
| Type C lesion (%) | 1916 (28.3%) | 260 (30.7%) | 34 (42.5%) | 0.008 |
| Three-vessel disease (%) | 1583 (22.6%) | 273 (30.7%) | 41 (47.1%) | <0.001 |
| Target LMT (%) | 195 (2.8%) | 37 (4.2%) | 4 (4.9%) | 0.042 |
| Target LAD (%) | 3415 (49.1%) | 367 (41.7%) | 35 (42.7%) | <0.001 |
| Cardio-protective medications |  |  |  |  |
| DAPT at arrival (%) | 5587 (79.8%) | 772 (81.3%) | 73 (83.9%) | 0.362 |
| DAPT at discharge (%) | 6742 (96.2%) | 840 (94.6%) | 83 (95.4%) | 0.056 |
| Beta-blocker at discharge (%) | 4976 (75.3%) | 613 (71.9%) | 54 (68.4%) | 0.038 |
| Statin at discharge (%) | 5936 (89.8%) | 706 (82.8%) | 58 (72.5%) | <0.001 |

CAD, coronary artery disease; CTO, chronic total occlusion; DAPT, dual antiplatelet therapy; IABP, intra-aortic balloon pump; LMT, left main trunk; LAD, left arterial descending.

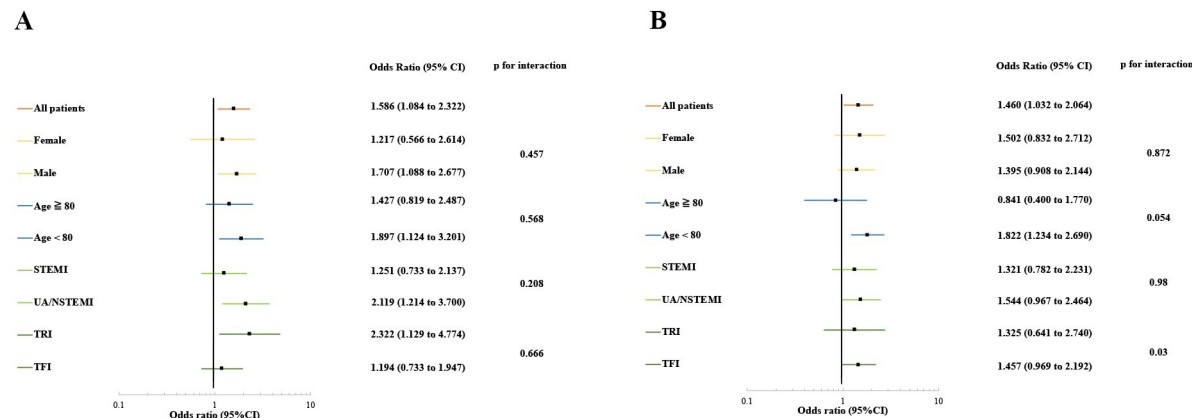

**Fig 3. Risk-adjusted outcomes and subgroup analysis across key subgroups.** Comparison of various in-hospital mortality rates (A) and bleeding complication rates (B) among patients with and without extra-cardiac lesion. Forest plots demonstrate comparative outcomes of acute coronary syndrome patients between those with and without extra-cardiac lesions. CI, confidence interval; NSTEMI, non-ST-elevation myocardial infarction; STEMI, ST-elevation myocardial infarction; TFI, transfemoral intervention; TRI, transradial intervention; UA, unstable angina.

between ECVD severity and procedural access site on in-hospital bleeding events on the multiplicative scale (OR: 2.105, 95% CI: 1.075–4.122), as well as on the additive scale (RERI: 0.669, 95% CI: −0.563–1.990) (Table 4).

## Temporal trends among patients with or without extra-cardiac vascular disease

S1 and S2 Tables present the temporal trend in the clinical and procedural demographics of patients with ECVD undergoing PCI between 2008 and 2017. The patients' age, sex, and baseline characteristics or proportion of complex PCIs did not change significantly over time. However, there was a considerable increase in the proportion of patients who received TRI, which quadrupled during the 9-year time period. In contrast, rates of TFI significantly decreased. With regard to medication, the rate of dual antiplatelet prescriptions upon arrival increased progressively. Furthermore, multivariate logistic regression analyses demonstrated that while the adjusted in-hospital mortality remained stable over time, the rate of bleeding complications declined significantly over the observed timespan (Tables 5 and 6 and S2 Fig). There was no significant decrease in deaths from 2008 through 2017, and the O/E mortality ratio remained unchanged (Table 5). In contrast, as illustrated in Table 6, there was a

**Table 3. Modification of the effect of extra-cardiac vascular disease severity on in-hospital mortality by procedural site.**

| | Without extra cardiac lesion | | With extra cardiac lesion | |
|---|---|---|---|---|
| | Deceased/Alive, n | OR (95% CI) | Deceased/Alive, n | OR (95% CI) |
| Radial Access | 27/2740 | 1.00 | 13/361 | 2.314 (1.160–4.616) p = 0.017 |
| Femoral Access | 95/4060 | 1.828 (1.175–2.844) p = 0.007 | 23/528 | 2.531 (1.441–4.444) p = 0.0012 |

Measure of interaction on additive scale: RERI = -0.617 (95% CI: -2.432 to 1.208); p = 0.51. Measure of interaction on multiplicative scale: OR: 0.864 (95% CI: 0.444 to 1.680); p = 0.666.

**Table 4. Modification of the effect of extra-cardiac vascular disease severity on in-hospital bleeding complication by procedural site.**

| | Without extra cardiac lesion | | With extra cardiac lesion | |
|---|---|---|---|---|
| | With/without bleeding complication, n | OR (95% CI) | With/without bleeding complication, n | OR (95% CI) |
| Radial Access | 48/2719 | 1.00 | 10/364 | 1.228 (0.612–2.466) p = 0.563 |
| Femoral Access | 151/4004 | 1.805 (1.293–2.521) p<0.001 | 36/515 | 2.702 (1.730–4.221) p<0.001 |

Measures of interaction on additive scale: RERI = 0.669 (95% CI: -0.563 to 1.900); p = 0.287. Measure of interaction on multiplicative scale: OR: 2.105 (95% CI: 1.075 to 4.122); p = 0.03.

significant improvement in the O/E bleeding event ratio in the most recent period (2015–2017). Although there was a slight decrease, the O/E mortality ratio remained greater than 2 (Table 5). On the contrary, the O/E bleeding complication ratio declined progressively to 0.16 at the end of the study (years 2015–2017) (Table 6). Moreover, this decrease in the bleeding complication rate was inversely correlated with the increase in TRI (S2 Fig). Compared to TFI, TRI was associated with a reduced in-hospital bleeding complication rate (OR: 0.479; 95% CI: 0.232–0.990); however, there was no association with in-hospital mortality (OR: 0.965; 95% CI: 0.450–2.068) (data not shown). Comparison of the trends in primary and secondary outcomes between patients with ACS with and without ECVD is shown in Fig 4. The difference in in-hospital mortality rate persisted between the two groups over time (Fig 4A). Although the bleeding complication rate among patients with ECVD was higher at the beginning of the study (2008–2010), it declined significantly in the latter part of the study period (2015–2017), to a rate that was below that of patients without ECVD (Fig 4B).

### Sensitivity and subgroup analyses

The results were unchanged after including patients who presented with cardiogenic shock or cardiopulmonary arrest with respect to the effect of concurrent ECVD on mortality (OR: 1.580; 95% CI: 1.212–2.061) and bleeding complications (OR: 1.397, 95% CI: 1.045–1.867) (S3–S5 Figs and S3 Table). Results of the subgroup analyses are presented in Fig 3. The interaction terms by access site (TRI/TFI) were significant in the adjusted analysis for bleeding complications, such that the effect of ECVD was more pronounced with TFI than with TRI.

## Discussion

### Key findings

Our study has three major findings. First, ACS patients with concurrent ECVD had worse in-hospital outcomes including mortality and bleeding complications compared to those without

**Table 5. Observed/expected ratios and adjusted odds ratios of in-hospital mortality.**

| Study years | Crude frequency (%) | Expected rates (%) | Observed to expected ratios | Adjusted OR (95% CI) |
|---|---|---|---|---|
| 2008 / 2010 | 4.4 | 1.3 | 3.38 | 1 |
| 2011 / 2014 | 4.1 | 1.7 | 2.41 | 0.814 (0.511, 1.299) |
| 2015 / 2017 | 3.8 | 1.6 | 2.37 | 0.776 (0.450, 1.341) |

CI, confidence interval; OR, odds ratio

**Table 6. Observed/Expected ratios and adjusted odds ratios of in-hospital bleeding complications.**

| Study years | Crude frequency (%) | Expected rates (%) | Observed to expected ratios | Adjusted or (95% CI) |
|---|---|---|---|---|
| 2008 / 2010 | 7.1 | 8.3 | 0.85 | 1 |
| 2011 / 2014 | 4.9 | 9.1 | 0.53 | 0.653 (0.345, 1.237) |
| 2015 / 2017 | 1.5 | 8.9 | 0.16 | 0.212 (0.048, 0.946) |

CI, confidence interval; OR, odds ratio

ECVDs. Second, although in-hospital mortality remained high among ECVD patients throughout the study period, the incidence of bleeding complications among patients with ECVD dropped significantly, owing to the increased use of TRI. Third, there was additive interaction between procedural access site and ECVD severity on bleeding events; the increased risk of bleeding associated with ECVD was more notable in patients undergoing PCI via the femoral artery. Our results add to the growing body of evidence that shows that the presence of ECVD is strongly associated with adverse in-hospital outcomes for patients with ACS undergoing PCI, even in the contemporary TRI-dominant PCI era. The current study is unique in that it integrated PAD and CVD and defined them as ECVD to assess the influence on various inpatient outcomes. We believe that this definition is more practical and may be easily utilized in clinical settings.

## Interpretation of results

In our study population, 5.4% of patients had PAD, which is similar to the incidence of PAD in previous reports from the British Cardiovascular Intervention Society database,[26] but lower than the incidence reported in the GRACE registry (5.7%).[5] On the contrary, patients included in this study were older, reflecting the rapidly aging Japanese society, which could have led to the higher prevalence of ECVD, particularly CVD. While there is a relatively low incidence of atherosclerotic disease other than cerebrovascular disease among East Asians compared to their Western counterparts, they are also known to be at higher risk of bleeding complications during PCI.[27] Indeed, in this study, patients with ECVD were at higher risk of bleeding compared to patients without ECVD. Currently, the biological mechanism for the increased risk of bleeding among ACS patients with ECVD is not clear. Achterberg et al. have

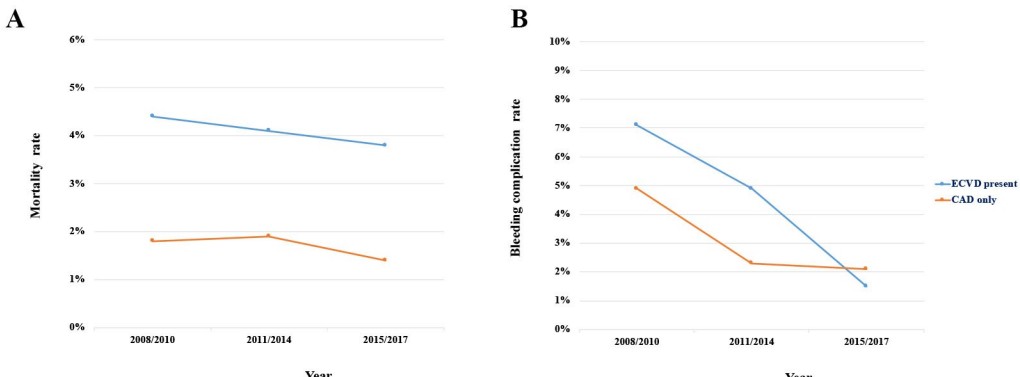

**Fig 4. Trends in primary and secondary outcomes among all patients with acute coronary syndrome stratified by the presence of extra-cardiac vascular disease.** Figures demonstrate (A) the in-hospital mortality and (B) bleeding complication trends among patients with acute coronary syndrome with or without extra-cardiac vascular disease, respectively. CAD, coronary artery disease; ECVD, extra-cardiac vascular disease.

hypothesized that the fragility of vessels, together with the heavy atherosclerotic burden of these patients, leads to rupture of the vessels and subsequent bleeding due to decreased vascular elasticity.[28] Steg et al. also reported a higher risk of bleeding among patients with PAD. [14] Peri-procedural bleeding is reported to be an indicator of subsequent major ischemic events and mortality.[29] On the contrary, to our surprise, the bleeding event rate did not increase in a stepwise fashion with the number of ECVDs; patients with multiple ECVDs had lower incidence of bleeding event than patients with only one ECVD. This could in part be explained by the decreased use of intra-aortic balloon pump, which is reported to increase bleeding events, among patients with multiple ECVDs. However, this could rather be reflecting a power issue, which we will address in the limitations section.

It is well documented that CAD patients with concomitant ECVD are less likely to receive optimal secondary prevention medical therapies, although they are likely to benefit from evidence-based therapies.[30] [31] This tendency was also observed in our study; that is, notably, the prescription of statin was the lowest in patients with both PAD and CVD. One possible explanation could be that patients with more extra-cardiac lesions are more likely intolerant to statins, presumably because the risk factors reported to be associated with statin intolerance coincide with the characteristics of patients with ECVD, such as advanced age, small body frame, frailty, or multisystem disease.[32]

Conventionally, performing PCI in patients with concurrent ECVD has been associated with low procedural success rates and high complication rates.[1, 5, 6] Our trend analyses demonstrated that contrary to the consistency observed in the mortality trend over the 9-year study period, bleeding complication rates among patients with ECVD dropped sharply in the most recent period (2015–2017). Although the underlying reasons for this improvement could not be determined, it may be attributable to the recent increase in TRI. Notably, increases in the use of TRI and of dual antiplatelet therapy were the only factors that changed during the study period. Clearly, an increased rate of dual antiplatelet therapy would likely promote an increased rate of bleeding complications; therefore, its use cannot be attributed to the decline in bleeding events. Additionally, our data confirm that TRI is associated with a reduced bleeding complication rate. This is supported by our results of the effect of ECVD on worsening bleeding complications, which was more pronounced among patients receiving PCI via TFI compared with TRI in multiplicative and additive interaction analyses. In other words, increase on TRI had a stronger effect in the reduction of bleeding complications. Moreover, our results demonstrate the progressive decline of the O/E bleeding event ratio, which further supports the effect of TRI on the reduction of bleeding complications. In general, global use of TRI has risen dramatically in recent years and has been proven to be accompanied by parallel improvements of clinical outcomes among patients with ACS.[18] Shoji. et al analyzed our JCD KiCS multicenter registry data to report the favorable effect of TRI to reduce periprocedural stroke.[33]

This study has several important clinical implications. Given the high prevalence of poor outcomes in patients with ACS and concurrent ECVD, there remains a need to develop new treatment strategies. Patients with ECVD could benefit from early initiation of prophylactic treatments before being admitted for ACS. In the randomized controlled Viborg Vascular trial, combined screening and intervention for abdominal aortic aneurysms, PAD, and hypertension were effective at reducing mortality.[34] Our findings suggest that increased adoption of TRI has led to reduced bleeding complication rates in recent years. Despite the recent increase in utilization of TRI, 20% of patients still received PCI via other access sites between 2015 and 2017. Furthermore, given the increased risk of bleeding among patients with concurrent ECVD, the choice of antiplatelet therapy is essential. Finally, given the recent improvements in clinical outcomes among patients with ECVD, PCI can be performed for these

patients relatively safely, and revascularization should not be avoided when it is deemed necessary.

### Strengths and limitations

Our study has several limitations. First, this was a retrospective study consisting of registry data; therefore, it is susceptible to selection bias. Second, we lacked precise information on the location of disease among patients with PAD or CVD. We cannot draw inferences about the specific effect of lower extremity artery disease or abdominal aortic disease among patients with PAD because the JCD-KiCS PCI registry does not capture the differences between these factors. Third, the decision to measure ABI was at the discretion of the attending physician, and, therefore, was not performed in all patients. This could have led to underestimation of the prevalence of PAD and potentially created information bias. Presumably, patients in more critical conditions could have been exempted from undergoing ABI testing. To address this, we performed subgroup and sensitivity analyses across different clinical presentation conditions, which revealed similar overall results to the main analyses. Fourth, the number of patients with two ECVDs was small and may not be representative of the population. This could have led to the unexpected lower incidence of in-hospital bleeding complications among patients with two ECVDs in comparison to those with one ECVD. Nevertheless, analysis of this subgroup was not the main purpose of our study. Our entire sample size, with 7,980 patients, was large enough to ensure the reliability of the study. Fifth, given the broad definitions of PAD and CVD in the study, our results may not be comparable with past studies. Nevertheless, the purpose of the current study was to assess the effect of the presence of any ECVD upon clinical outcomes, and inclusion of various types of extra-cardiac lesions was necessary.

### Conclusions

In an era in which the importance of extra-cardiac lesions is increasingly recognized, this study adds to the growing body of evidence of the prognostic significance of ECVD, inclusive of PAD or CVD, as a risk factor for adverse inpatient outcomes after PCI to treat ACS. Our data clarify the recent reduction in the magnitude of the effect of ECVD on the risk of adverse inpatient events, which can be attributed to the widespread use of TRI.

### Supporting information

**S1 Fig. In-hospital outcomes according to the extent of vascular disease.** Bar chart demonstrating the rates of in-hospital deaths **(A)** and bleeding complications **(B)** in the three groups. ACS, acute coronary syndrome; CAD, coronary artery disease.
(TIF)

**S2 Fig.** (A) Trends of in-hospital outcomes for patients with ACS and concomitant ECVDs in relation to the percentage of transradial intervention. Forest plots illustrate comparative outcomes with reference to the most recent years (2008–2010).
**(B)** Trends in transradial intervention rate stratified by the presence of extra-cardiac vascular disease. Figure demonstrate the rate of transradial coronary intervention rate with or without extra-cardiac vascular disease, respectively. ACS, acute coronary syndrome; ECVD, extracardiac vascular disease; TRI, transradial intervention.
(TIF)

**S3 Fig. In-hospital outcomes according to the extent of vascular disease when patients with cardiogenic shock and cardiopulmonary arrest were included.** Bar chart demonstrating the rates of in-hospital deaths (A) and bleeding complications (B) in the three groups.

ACS, acute coronary syndrome; CAD, coronary artery disease.
(TIFF)

**S4 Fig. Risk-adjusted outcomes and subgroup analysis across key subgroups when patients with cardiogenic shock and cardiopulmonary arrest were included.** Comparison of various in-hospital mortality rates (A) and bleeding complication rates (B) among patients with and without extra-cardiac lesion. Forest plots demonstrate comparative outcomes of acute coronary syndrome patients between those with and without extra-cardiac lesions.
CI, confidence interval; NSTEMI, non-ST-elevation myocardial infarction; STEMI, ST-elevation myocardial infarction; TFI, transfemoral intervention.
TRI, transradial intervention; UA, unstable angina.
(TIF)

**S5 Fig. Trends in primary and secondary outcomes among all patients with ACS stratified by the presence of ECVD when patients with cardiogenic shock and cardiopulmonary arrest were included.** Figures demonstrate the in-hospital mortality (A) and bleeding complication (B) trends among patients with ACS with or without ECVD, respectively.
ACS, acute coronary syndrome; CAD, coronary artery disease; ECVD, extracardiac vascular disease.
(TIF)

**S1 Table. Trends in clinical characteristics among patients with extra-cardiac lesion.** BMI, body mass index; CAD, coronary artery disease; CABG, coronary artery bypass grafting; MI, myocardial infarction; NSTEMI, non-ST-elevation myocardial infarction; PCI, percutaneous coronary intervention; STEMI, ST-elevation myocardial infarction.
(DOCX)

**S2 Table. Trends in procedural and prescription characteristics among patients with extra-cardiac lesion.** CTO, chronic total occlusion; DAPT, dual antiplatelet therapy; IABP, intra-aortic balloon pump; LAD, left arterial descending; LMT, left main trunk.
(DOCX)

**S3 Table.** Modification of the effect of extra-cardiac vascular disease severity on in-hospital mortality **(A)** and bleeding complication **(B)** by procedural site when patients with cardiogenic shock and cardiopulmonary arrest were included.
(A) Measure of interaction on additive scale: RERI (relative excess risk due to interaction) = -0.0528 (95% CI: -1.354 to 1.248); p = 0.9366. Measure of interaction on multiplicative scale: OR: 0.680 (95% CI: 0.399 to 1.158); p = 0.156.
(B) Measures of interaction on additive scale: RERI = 0.383 (95% CI: -0.723 to 1.489);
p = 0.497. Measure of interaction on multiplicative scale: OR: 2.005 (95% CI: 1.511 to 2.661); p<0.001.
(DOCX)

## Acknowledgments

The authors thank all the investigators, clinical coordinators, and institutions involved in the JCD- KiCS.

**Investigators:** Yohei Numasawa, Toshiki Kuno, Makoto Tanaka(Japanese Red Cross Ashikaga Hospital), Yutaka Okada (Eiju General Hospital), Soushin Inoue, Iwao Nakamura (Hino Municipal Hospital), Takaharu Katayama, Shunsuke Takagi, Takashi Matsubara (Hiratsuka City Hospital), Masashi Takahashi, Keishu Li, Koichiro Sueyoshi (Kawasaki City Municipal

Hospital), Atsushi Anzai, Kentaro Hayashida, Takashi Kawakami, Hideaki Kanazawa, Shunsuke Yuasa, Yuichiro Maekawa (Keio University School of Medicine), Masahiro Suzuki, Keisuke Matsumura (National Hospital Organization Saitama National Hospital) Ryota Tabei, Yukinori Ikegami, Jun Fuse, Munehisa Sakamoto, Yukihiko Momiyama (National Hospital Organization Tokyo Medical Center), Ayaka Endo, Tasuku Hasegawa, Toshiyuki Takahashi, Susumu Nakagawa (Saiseikai Central Hospital), Fumiaki Yashima, Koji Ueno, Kenichiro Shimoji, Shigetaka Noma (Saiseikai Utsunomiya Hospital), Masahito Munakata, Takashi Akima, Shiro Ishikawa, Takashi Koyama (Saitama City Hospital), Atsushi Mizuno (St Luke's International Hospital Heart Center), Toshimi Kageyama, Kazunori Moritani, Masaru Shibata (Tachikawa Kyosai Hospital), Hiroaki Sukegawa, Yoshinori Mano, Takahiro Oki (Tokyo Dental College Ichikawa General Hospital), Daisuke Shinmura, Koji Negishi, and Takahiro Koura (Yokohama Municipal Hospital)

**Clinical Coordinators:** Junko Susa, Ayano Amagawa, Hiroaki Nagayama, Miho Umemura, Itsuka Saito, and Ikuko Ueda

## Author Contributions

**Conceptualization:** Masaki Kodaira, Mitsuaki Sawano, Shun Kohsaka.

**Data curation:** Masaki Kodaira, Mitsuaki Sawano.

**Formal analysis:** Masaki Kodaira.

**Funding acquisition:** Mitsuaki Sawano, Shun Kohsaka.

**Investigation:** Masaki Kodaira, Mitsuaki Sawano, Shun Kohsaka.

**Methodology:** Masaki Kodaira.

**Project administration:** Masaki Kodaira.

**Supervision:** Yohei Numasawa.

**Visualization:** Masaki Kodaira.

**Writing – original draft:** Masaki Kodaira.

**Writing – review & editing:** Masaki Kodaira, Mitsuaki Sawano, Toshiki Kuno, Yohei Numasawa, Shigetaka Noma, Masahiro Suzuki, Shohei Imaeda, Ikuko Ueda, Keiichi Fukuda, Shun Kohsaka.

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
