## [Decision Letter · Decision Letter 0]

1 Aug 2019

PONE-D-19-17185

Outcomes of acute coronary syndrome patients with concurrent extra-cardiac vascular disease in the era of transradial coronary intervention: a retrospective multicentre cohort study

PLOS ONE

Dear Dr. Kodaira,

Thank you for submitting your manuscript to PLOS ONE. After careful consideration, we feel that it has merit but does not fully meet PLOS ONE’s publication criteria as it currently stands. Therefore, we invite you to submit a revised version of the manuscript that addresses the points raised during the review process.

We would appreciate receiving your revised manuscript by Sep 15 2019 11:59PM. To enhance the reproducibility of your results, we recommend that if applicable you deposit your laboratory protocols in protocols.io, where a protocol can be assigned its own identifier (DOI) such that it can be cited independently in the future. For instructions see: http://journals.plos.org/plosone/s/submission-guidelines#loc-laboratory-protocols

We look forward to receiving your revised manuscript.

Kind regards,

Corstiaan den Uil

Academic Editor

PLOS ONE

Journal Requirements:

[ have read the journal's policy and the authors of this manuscript have the following competing interests:

Dr. Kohsaka has received grants from Bayer Yakuhin and Daiichi-Sankyo; has received lecture fees from Bayer Yakuhin and Bristol-Myers Squibb.]. 

We note that you received funding from a commercial source: [Bayer and Bristol-Myers Squibb]

Reviewers' comments:

Reviewer's Responses to Questions

**Comments to the Author**

1. Is the manuscript technically sound, and do the data support the conclusions?

Reviewer #1: Partly

2. Has the statistical analysis been performed appropriately and rigorously? 

Reviewer #1: Yes

3. Have the authors made all data underlying the findings in their manuscript fully available?

Reviewer #1: Yes

4. Is the manuscript presented in an intelligible fashion and written in standard English?

Reviewer #1: Yes

5. Review Comments to the Author

Reviewer #1: I read with interest the manuscript by Kodaira et al. In this study, the authors used data from a large multicenter registry to examine clinical outcomes in patients with ECVD undergoing PCI for ACS. The main objectives of the study were: 1) to examine the impact of ECVD on in-hospital outcomes in patients with ACS, and 2) determine whether access site (radial versus femoral) has a modulating effect on the association between ECVD and in hospital outcomes.

As the authors mention, the MATRIX trial suggested clinical benefits with radial compared to femoral access in patients with ACS. In MATRIX, there was no significant interaction between the presence of PVD and the treatment effect of transradial access. Furthermore, the results of SAFARI (presented at ACC 2019 but not yet published) did not demonstrate a difference in clinical outcomes (including bleeding) with radial versus femoral access in patients undergoing primary PCI for STEMI.

The authors should be commended for their work. Their analysis is based on a large dataset and addresses an important clinical question. In addition, manuscript is well written with meticulous methodology. However, as with any observational study, there are important limitations. In particular, inferring casualty from a non-randomized comparison.

Comments/suggestion:

Methods

1) For multivariate logistic regression, how were explanatory variables chosen?

2) Do the authors have data on additional clinical outcomes (e.g. stroke, myocardial infarction, etc)? Some interventionalists worry about radial access in patients with cerebrovascular disease due to concerns about stroke. Additional data would be informative if available.

Results

1) As a reader, I found differences in pharmacotherapy in relation to ECVD interesting (Table 2). These may be reflect differences in practice patterns, which may not have been captured in multivariate logistic regression. Addressing these in the discussion would be of value.

2) The trends highlighted in figure 4b are interesting. Between 2011/2014 and 2015/2017, there appears to be a disproportionate decrease in bleeding complications in patients with versus without ECVD. Do the authors have a hypothesis as to why this occurred? If the hypothesis is changes in TRI frequency, was there a concomitant disproportionate increase in TRI during the same period in patients with versus without ECVD?

3) In-hospital bleeding in relation to ECVD (as highlighted in fig S1B) suggests that bleeding is higher with 1 ECVD as opposed to 2 ECVD (which has the lowest observed bleeding rate among all groups). Do the authors have a potential explanation for this observation? Based on the authors’ hypothesis, you should expect similar or more bleeding in patients with more ECVD.

4) In Fig S4B, p value for interaction for radial access is <0.001 even though the odds ratios point estimates are very similar. Could there be an error?

Discussion

1) In line 432, the authors suggest favoring clopidogrel versus ticagrelor to avoid bleeding events. In my opinion, the reference provided and the observations in the current analysis do not provide enough substrate for this recommendation. I would consider removing this statement.

Conclusion

1) Both statements in the conclusion (TRI has reduced bleeding and TRI should be used in patients with ECVD) are too strong as they imply causality. Suggest a more conservative conclusion highlighting observed associations only.

Edits

1) In line 109, “fulfill” is misspelled

2) As a reader, in line 206, it was not readily apparent to me that the third type of vascular disease refers to CAD. I would suggest rewording to avoid confusion.

6. PLOS authors have the option to publish the peer review history of their article (what does this mean?). If published, this will include your full peer review and any attached files.

Reviewer #1: No

---

## [Author Response · Author response to Decision Letter 0]

14 Sep 2019

14 September, 2019

Dr. Corstiaan Den Uil

Academic Editor

PLoS ONE

Re: Resubmission of manuscript ID: PONE-D-19-17185

Dear Dr. Uil,

Please find enclosed our manuscript titled, “Outcomes of acute coronary syndrome patients with concurrent extra-cardiac vascular disease in the era of transradial coronary intervention: a retrospective multicenter cohort study,” which we are resubmitting to PLoS ONE.

We thank you and the reviewers for your thoughtful suggestions and insights. We believe that the comments allowed us to improve the quality of our manuscript. Our point-by-point responses to the comments are shown below. Changes to our manuscript are highlighted in yellow.

Please let us know if any further clarification is required. We trust that our manuscript is now suitable for publication in the PLoS ONE. We look forward to hearing from you.

Yours sincerely,

Masaki Kodaira, M.D. Ph.D.

Department of Cardiology, Japanese Red Cross Ashikaga Hospital

284-1 Yobe-cho, Ashikaga, Tochigi 326-0843, Japan

Tel: +81-284-21-0121

Fax: +81-284-21-6810　

E-mail: m.kodaira@ashikaga.jrc.or.jp

\f

RESPONSE TO COMMENTS

We have formatted with these requirements. If something is wrong, please let us know.

Data are available only upon request according to the “Act on the Protection of

Personal Information” Law (as of May 2017) and the “Ethical Guidelines for Medical and Health Research Involving Human Subjects” (as of March 2015). The current study data was obtained from the JCD-KiCS PCI registry and would be available upon request to the author Dr. Shun Kohsaka MD (E-mail: sk@keio.jp).

[ have read the journal's policy and the authors of this manuscript have the following competing interests: Dr. Kohsaka has received grants from Bayer Yakuhin and Daiichi-Sankyo; has received lecture fees from Bayer Yakuhin and Bristol-Myers Squibb.]. 

We note that you received funding from a commercial source: [Bayer and Bristol-Myers Squibb]

Dr. Kohsaka has received grants from Bayer Yakuhin and Daiichi-Sankyo; has received lecture fees from Bayer Y.akuhin and Bristol-Myers Squibb. This does not alter our adherence to PLOS ONE policies on sharing data and materials.

Reviewer #1: I read with interest the manuscript by Kodaira et al. In this study, the authors used data from a large multicenter registry to examine clinical outcomes in patients with ECVD undergoing PCI for ACS. The main objectives of the study were: 1) to examine the impact of ECVD on in-hospital outcomes in patients with ACS, and 2) determine whether access site (radial versus femoral) has a modulating effect on the association between ECVD and in hospital outcomes.  As the authors mention, the MATRIX trial suggested clinical benefits with radial compared to femoral access in patients with ACS. In MATRIX, there was no significant interaction between the presence of PVD and the treatment effect of transradial access. Furthermore, the results of SAFARI (presented at ACC 2019 but not yet published) did not demonstrate a difference in clinical outcomes (including bleeding) with radial versus femoral access in patients undergoing primary PCI for STEMI.  The authors should be commended for their work. Their analysis is based on a large dataset and addresses an important clinical question. In addition, manuscript is well written with meticulous methodology. However, as with any observational study, there are important limitations. In particular, inferring casualty from a non-randomized comparison.

Comments/suggestion:  Methods 1) For multivariate logistic regression, how were explanatory variables chosen?

As described in lines 161–163 of the Methods section, our multivariate logistic regression analysis was performed using explanatory variables which were incorporated in the National Cardiovascular Data Registry (NCDR) risk models for in-hospital mortality (1) and bleeding risk.(2) 

As for its details, the NCDR for catheterization (CathPCI) registry provides the ideal infrastructure to derive procedure risk models with more than 1500 participating centers, which is co-sponsored by the American College of Cardiology and the Society for Cardiovascular Angiography and Intervention (http://www.ncdr.com) In addition, Peterson et al. developed the relevant risk model for 30-day PCI mortality by analyzing 600,533 consecutive PCI admissions between January 2004 and March 2007 recorded in the NCDR CathPCI registry.(1) The risk model was applied to prospective validation sample sets to demonstrate excellent discrimination (c-index: 0.91). Moreover, Rao et al. identified factors associated with major complications occurring within 72 hours after PCI by analyzing 1,043,759 PCI procedures in the NCDR CathPCI registry performed between February 2008 and April 2011.(2) The model also had good discrimination in the validation sample with c-index of 0.77. 

2) Do the authors have data on additional clinical outcomes (e.g. stroke, myocardial infarction, etc.)? Some interventionists worry about radial access in patients with cerebrovascular disease due to concerns about stroke. Additional data would be informative if available.

 

Thank you for your interest in additional clinical outcomes. Yes, we have data on these clinical outcomes. 

Recently, we examined the effect of TRI on periprocedural stroke by analyzing 17,966 patients who underwent PCI between 2008 and 2016 in our JCD KiCS registry.(3) Within this paper, multivariable logistic regression analysis (odds ratio; 0.33; 95% CI, 0.16–0.71; p = 0.004) and propensity score matching analysis (0.1% versus 0.4%; p = 0.014) revealed that TRI, in comparison with TFI, was associated with lower risk of periprocedural stroke. Shoji et al. attributed this favorable effect of TRI to its less catheter contact with the aortic arch (4) and avoidance of contact with the abdominal or descending thoracic portions of the aorta.(5)

As regards to periprocedural myocardial infarction, analysis of our current study population demonstrated that patients who underwent TRI had significantly lower incidence of periprocedural myocardial infarction than those who underwent TFI (0.89% versus 1.78%, p = 0.001). However, we decided not to include this result in our revised manuscript, because cardiac biomarker testing is not performed routinely after PCI. Arai et al. reported from our JCD KiCS registry that only 25.2% of the patients received cardiac biomarker assessment after PCI.(6) This low rate of cardiac biomarker assessment was also reported in the United States, at 24.7%.(7)

Please find the following sentences added to the revised manuscript on Shoji’s work on the favorable effect of TRI on the reduction of periprocedural stroke after PCI. 

Discussion section (page 34, lines 421–422):

Shoji. et al analyzed our JCD KiCS multicenter registry data to report the favorable effect of TRI to reduce periprocedural stroke. 

Reference 33 (Circ Cardiovasc Interv. 2018;11(12):e006761) was added to support this. 

 Results  1) As a reader, I found differences in pharmacotherapy in relation to ECVD interesting (Table 2). These may reflect differences in practice patterns, which may not have been captured in multivariate logistic regression. Addressing these in the discussion would be of value.

 

Thank you for having interest in our data on pharmacotherapy. The prescription rate of cardio-protective drugs recommended in the guidelines, statin, and beta-blockers were significantly lower in those with concomitant ECVD than in those without ECVD. This finding had been observed in past reports. We have added the following paragraph and added references 30–32. 

Discussion section (page 32, lines 391–399):

It is well documented that CAD patients with concomitant ECVD are less likely to receive optimal secondary prevention medical therapies, although they are likely to benefit from evidence-based therapies. This tendency was also observed in our study; that is, notably the prescription of statin was the lowest in patients with both PAD and CVD. One possible explanation could be that patients with more extra-cardiac lesions are more likely intolerant to statins, presumably because the risk factors reported to be associated with statin intolerance coincide with the characteristics of patients with ECVD, such as advanced age, small body frame, frailty, or multisystem disease.

2) The trends highlighted in figure 4b are interesting. Between 2011/2014 and 2015/2017, there appears to be a disproportionate decrease in bleeding complications in patients with versus without ECVD. Do the authors have a hypothesis as to why this occurred? If the hypothesis is changes in TRI frequency, was there a concomitant disproportionate increase in TRI during the same period in patients with versus without ECVD?

Thank you for commenting on the most essential finding of our study. As we stated in page 34, lines 422–435, our hypothesis is that increase in transradial intervention (TRI) during the period led to the disproportionate decrease in bleeding complications in ACS patients with ECVD versus without ECVD. However, this increase in TRI was not disproportionate between patients with and without ECVD. We have added supplemental figure (S2B Fig) to demonstrate this. In addition, we have edited Supplemental Figure S2A to include the rate of TRI according to the presence of ECVD. Although its increase during this period was proportionate, increase in TRI may play a role by causing disproportionate decrease in bleeding complications between patients with and without ECVD. Both our multiplicative and additive interaction analyses supported that the effect of ECVD on worsening bleeding complications was more pronounced among ACS patients undergoing transfemoral intervention (TFI) compared with those undergoing TRI (page 33 lines 412–414). In other words, increase in TRI (=decrease in TFI) had a stronger effect in the reduction of bleeding complications among ACS patients with ECVD compared to those without ECVD. 

The following sentences from our original manuscript should clarify the points:

Discussion section (page 33, lines 403–415)

Our trend analyses demonstrated that contrary to the consistency observed in the mortality trend over the 9-year study period, bleeding complication rates among patients with ECVD dropped sharply in the most recent period (2015–2017). Although the underlying reasons for this improvement could not be determined, it may be attributable to the recent increase in TRI. Notably, increases in the use of TRI and of dual antiplatelet therapy were the only factors that changed during the study period. Clearly, an increased rate of dual antiplatelet therapy would likely promote an increased rate of bleeding complications; therefore, its use cannot be attributed to the decline in bleeding events. Additionally, our data confirm that TRI is associated with a reduced bleeding complication rate. This is supported by our results of the effect of ECVD on worsening bleeding complications, which was more pronounced among patients receiving PCI via TFI compared with TRI in multiplicative and additive interaction analyses.

As our explanation was not clear, we added the following sentence to the Discussion section in our revised manuscript. 

Discussion section (page 33, lines 415–416)

In other words, increase in TRI had a stronger effect in the reduction of bleeding complications among patients with compared to those without ECVD. 

 3) In-hospital bleeding in relation to ECVD (as highlighted in fig S1B) suggests that bleeding is higher with 1 ECVD as opposed to 2 ECVD (which has the lowest observed bleeding rate among all groups). Do the authors have a potential explanation for this observation? Based on the authors’ hypothesis, you should expect similar or more bleeding in patients with more ECVD.

We thank the reviewer for the comments and deep insights. We were also surprised to find lower incidence of in-hospital bleeding complications among patients with multiple ECVDs. This result is difficult to interpret because it conflicts with what can be expected from the unfavorable baseline characteristics shown in Table 1; patients with multiple ECVDs were the oldest, had the lowest body mass index, most frequently had past history of PCI, and possessed chronic kidney disease. All these variables are reported to be associated with higher incidence of in-hospital bleeding complications. 

On the other hand, Table 2 provides another interesting aspect on the procedural characteristic of patients with multiple ECVDs. Notably, these patients had significantly lower incidence of intra-aortic balloon pump (IABP) insertion (4.6%) compared those with only one ECVD (8.9%) or CAD only (7.7%). Lee et al. demonstrated in their meta-analysis that the use of IABP, which requires insertion of a relatively large size sheath via the femoral artery, was associated with increased risk of moderate to severe bleeding compared with medical therapy (relative risk 1.41, 95% CI 1.01–2.08).(8) Although the reason for this difference in the rate of IABP use is not clear, we speculate that the physicians avoided using IABP for patients with two ECVDs, considering their unfavorable condition. 

Finally, most of all, we recognize that this is more of a power issue. The number of patients with two ECVDs was small (n = 87) and may not be representative of the population. In other words, the small sample size of this subpopulation with multiple ECVDs subjects its results to sampling variation and random error.(9) Nevertheless, we would like to underline that our study population as a whole had a large sample size of 7,980 patients to ensure the integrity of the results. We, therefore, added the following sentences below. 

Discussion section (page 31, lines 382 to page 32, line 388):

On the contrary, to our surprise, the bleeding event rate did not increase in a stepwise fashion with the number of ECVDs; patients with multiple ECVDs had lower incidence of bleeding event than patients with only one ECVD. This could in part be explained by the decreased use of intra-aortic balloon pump, which is reported to increase bleeding events, among patients with multiple ECVDs. However, this could rather be reflecting a power issue, which we will address in the limitations section.

Discussion, Strength and limitations section (page 36, lines 453–459):

Fourth, the sample size of patients with multiple ECVDs was small and may not be representative of the population. This could have led to the unexpected lower incidence of in-hospital bleeding complications among patients with multiple ECVDs in comparison to those with one ECVD. Nevertheless, analysis of this subgroup was not the main purpose of our study. Our entire sample size, with 7,980 patients, was large enough to ensure the reliability of the study.

 4) In Fig S4B, p value for interaction for radial access is <0.001 even though the odds ratios point estimates are very similar. Could there be an error?

Thank you for bringing this to our attention. We reexamined our analysis for Fig S4B to find an error in our calculation for the p value of interaction for radial access. It should be 0.718, and we have edited Fig S4B accordingly. We apologize for the miscalculation. We also performed recalculation for other analyses to make sure that other values were correct.

  Discussion 1) In line 432, the authors suggest favoring clopidogrel versus ticagrelor to avoid bleeding events. In my opinion, the reference provided and the observations in the current analysis do not provide enough substrate for this recommendation. I would consider removing this statement.

Thank you for pointing out. We agree with the reviewer that there is not enough evidence to support this statement, and the sentence has been omitted accordingly. 

  Conclusion  1) Both statements in the conclusion (TRI has reduced bleeding and TRI should be used in patients with ECVD) are too strong as they imply causality. Suggest a more conservative conclusion highlighting observed associations only.

We agree with the reviewer that our statements in the conclusion of the abstract were rather strong. In our revised manuscript, our conclusion in the abstract is now in lined with the conclusion of the main text. 

Abstract conclusions (page 4, lines 46–49):

Overall, the presence of ECVD was a risk factor for adverse outcomes after PCI for ACS, both mortality and bleeding complications. In the most recent years, the incidence of bleeding complication among patients with ECVD decreased significantly coinciding with the rapid increase of TRI. 

  Edits  1) In line 109, “fulfill” is misspelled.

PLoS One’s submission guideline plainly states “English” as the preferred language and does not indicate whether British English or US English is preferred. We, therefore, submitted our initial manuscript in British English, including “fulfill.” After acknowledging that the reviewer prefers US English, we have edited the manuscript, including the title. Please find below the changes made from British English to US English.

Title (page 1, line 3), Abstract (page 3, line 31):

multicentre > multicenter

Methods and materials section (page 8, line 109):

fulfil > fulfill

Methods and materials section (page 11, line 162) and (page 13, line 187):

haemoglobin > hemoglobin

Discussion section (page 31, line 377):

hypothesised > hypothesized

 2) As a reader, in line 206, it was not readily apparent to me that the third type of vascular disease refers to CAD. I would suggest rewording to avoid confusion.

We apologize for the unintelligibility. Taking the reviewer’s advice, we have modified the sentence as shown below to make it clear to the readers that the third type was CAD.

Results section (page 14, lines 201–204):

The current analysis included 7,980 patients who underwent PCI for ACS. Of these, 888 patients (11.1%) had one additional vascular disease (CAD + PAD: 345 patients [4.3%], CAD + CVD: 543 patients [6.8%]), whereas 87 patients (1.1%) had all three types (CAD + PAD + CVD) of vascular disease (Fig 2).

References for the rebuttal letter

1. Peterson ED, Dai D, DeLong ER, Brennan JM, Singh M, Rao SV, et al. Contemporary mortality risk prediction for percutaneous coronary intervention: results from 588,398 procedures in the National Cardiovascular Data Registry. J Am Coll Cardiol. 2010;55(18):1923-32.

2. Rao SV, McCoy LA, Spertus JA, Krone RJ, Singh M, Fitzgerald S, et al. An updated bleeding model to predict the risk of post-procedure bleeding among patients undergoing percutaneous coronary intervention: a report using an expanded bleeding definition from the National Cardiovascular Data Registry CathPCI Registry. JACC Cardiovascular interventions. 2013;6(9):897-904.

3. Shoji S, Kohsaka S, Kumamaru H, Sawano M, Shiraishi Y, Ueda I, et al. Stroke after percutaneous coronary intervention in the era of transradial intervention. Circ Cardiovasc Interv. 2018;11(12):e006761.

4. Khoury Z, Gottlieb S, Stern S, Keren A. Frequency and distribution of atherosclerotic plaques in the thoracic aorta as determined by transesophageal echocardiography in patients with coronary artery disease. Am J Cardiol. 1997;79(1):23-7.

5. Roberts WC, Schussler JM. Frequency of plaque dislodgement and embolization in transradial vs transfemoral approaches for left-sided cardiac catheterization: clinically silent vs clinically apparent embolism. JAMA Cardiol. 2018;3(7):551-2.

6. Arai T, Yuasa S, Miyata H, Kawamura A, Maekawa Y, Ishikawa S, et al. Incidence of periprocedural myocardial infarction and cardiac biomarker testing after percutaneous coronary intervention in Japan: results from a multicenter registry. Heart Vessels. 2013;28(6):714-9.

7. Wang TY, Peterson ED, Dai D, Anderson HV, Rao SV, Brindis RG, et al. Patterns of cardiac marker surveillance after elective percutaneous coronary intervention and implications for the use of periprocedural myocardial infarction as a quality metric: a report from the National Cardiovascular Data Registry (NCDR). J Am Coll Cardiol. 2008;51(21):2068-74.

8. Lee JM, Park J, Kang J, Jeon KH, Jung JH, Lee SE, et al. The efficacy and safety of mechanical hemodynamic support in patients undergoing high-risk percutaneous coronary intervention with or without cardiogenic shock: Bayesian approach network meta-analysis of 13 randomized controlled trials. Int J Cardiol. 2015;184:36-46.

9. Button KS, Ioannidis JP, Mokrysz C, Nosek BA, Flint J, Robinson ES, et al. Power failure: why small sample size undermines the reliability of neuroscience. Nat Rev Neurosci. 2013;14(5):365-76.

---

## [Editor Report · Decision Letter 1]

17 Sep 2019

Outcomes of acute coronary syndrome patients with concurrent extra-cardiac vascular disease in the era of transradial coronary intervention: a retrospective multicenter cohort study

PONE-D-19-17185R1

Dear Dr. Kodaira,

We are pleased to inform you that your manuscript has been judged scientifically suitable for publication and will be formally accepted for publication once it complies with all outstanding technical requirements.

With kind regards,

Corstiaan den Uil

Academic Editor

PLOS ONE
---

## [Editor Report · Acceptance letter]

27 Sep 2019

PONE-D-19-17185R1 

Outcomes of acute coronary syndrome patients with concurrent extra-cardiac vascular disease in the era of transradial coronary intervention: a retrospective multicenter cohort study 

Dear Dr. Kodaira:

I am pleased to inform you that your manuscript has been deemed suitable for publication in PLOS ONE. Congratulations! Your manuscript is now with our production department. 

With kind regards,

on behalf of

Dr. Corstiaan den Uil 

Academic Editor

PLOS ONE